# SYNTHETIC DATA AS VALIDATION

## ABSTRACT

This study leverages synthetic data as a validation set to reduce overfitting and ease the selection of the best model in AI development. While synthetic data have been used for augmenting the training set, we find that synthetic data can also significantly diversify the validation set, offering marked advantages in domains like healthcare, where data are typically limited, sensitive, and from out-domain sources (i.e., hospitals). In this study, we illustrate the effectiveness of synthetic data for early cancer detection in computed tomography (CT) volumes, where synthetic tumors are generated and superimposed onto healthy organs, thereby creating an extensive dataset for rigorous validation. Using synthetic data as validation can improve AI robustness in both in-domain and out-domain test sets. Furthermore, we establish a new continual learning framework that continuously trains AI models on a stream of out-domain data with synthetic tumors. The AI model trained and validated in dynamically expanding synthetic data can consistently outperform models trained and validated exclusively on real-world data. Specifically, the DSC score of liver tumor segmentation improves from 26.7% (95% CI: 22.6%–30.9%) to 34.5% (30.8%–38.2%) when evaluated on an in-domain dataset and from 31.1% (26.0%–36.2%) to 35.4% (32.1%–38.7%) on an out-domain dataset. Importantly, the performance gain is particularly significant in identifying very tiny liver tumors (radius $< 5$mm) in CT volumes, with Sensitivity improving from 33.1% to 55.4% on an in-domain dataset and 33.9% to 52.3% on an out-domain dataset, justifying the efficacy in the early detection of cancer. The application of synthetic data, from both training and validation perspectives, underlines a promising avenue to enhance AI model robustness when dealing with data from varying domains. Our code is attached as supplementary and will be publicly available.

## 1 INTRODUCTION

Standard AI development divides the dataset into a training set and a test set; the former is used for model training, and the latter is for evaluation (Russell, 2010; Gareth et al., 2013). The AI model is updated every *training epochs*, resulting in a number of intermediate models during the training trajectory. The performance of these models tends to improve on the training set, but this does not mean the performance on the test set also improves due to the over-fitting problem (Kuhn et al., 2013). A question then arises: *How do we identify the best model that performs well on the test set, especially when it is evaluated on test sets taken from different domains?* A prevalent strategy is to delineate a validation set from the training set (Ripley, 2007). This validation set neither contributes to training nor to evaluating the AI performance. Instead, it functions as an independent set to fix the training hyper-parameters and, more importantly, to estimate the performance of each model on different datasets, thus enabling the selection of the best model from the many intermediate models during the training trajectory.

The validation set is often kept small. Naturally, we would like to maximize the use of the training data. Annotating data for AI training is time-consuming and expensive, requiring specialized expertise, so the annotated datasets are limited in size in many fields (Zhou, 2021). Allocating too many annotated data for validation would inevitably diminish the training set size and compromise the AI training. On the other hand, the validation set should be sufficiently representative to provide a reliable performance estimate on unseen data. An overly small validation set might risk the reliability of performance estimation and checkpoint selection. As a result, the calibration of the validation set remains largely empirical and lacks systematic investigation for better alternatives to select the best

checkpoint. Fulfilling this knowledge gap is particularly important in scenarios where real-world data are scarce, sensitive, or costly to collect and annotate, as seen in the field of *AI for healthcare* (Zhou et al., 2022). Therefore, our study uses early detection of cancerous tumors in computed tomography (CT) volumes as a demonstration. While early detection of cancer holds immense clinical potential, it faces profound constraints like disease prevalence and annotation difficulty to collect examples of early-stage tumors (Crosby et al., 2022). The scarcity of annotated early cancer not only constrains the data available for validation but also amplifies the overfitting problem inherent in a small, biased validation set, potentially causing underdiagnosis and overdiagnosis.

We propose using synthetic data as validation, a strategy that guarantees the full utilization of the training set while ensuring ample data diversity for validation. Data synthesis has held longstanding interest and presents numerous intriguing merits for augmenting training and test data (Hu et al., 2023; Gao et al., 2023) as reviewed in §2, but its use in validation has seldom been explored. We find that synthetic data can facilitate a more reliable performance estimate on unseen data and effectively address the constraints commonly associated with small, biased validation sets. Specifically, we synthesize tumors in the healthy liver, which gives us orders of magnitude larger datasets for training. To ensure the realism of the synthetic tumors, we employ a modeling-based strategy (Hu et al., 2023) to simulate cancerous tumors with controlled shape, size, texture, location, and intensity. The use of diverse, healthy CT volumes, supplemented with synthetic tumors, as validation has demonstrated efficacy in mitigating model overfitting and enhancing the selection of checkpoints. Furthermore, we relieve the pressing demand for human annotations to train AI models by utilizing CT volumes with synthetic tumors as the training set. We then assess the model's performance using a substantial number of publicly available, fully-annotated CT volumes with real-world cancerous tumors, showing that our models generalize well to these volumes from different hospitals and accurately segment the tumors at their early stage. Our findings can be summarized as follows:

1. The best model checkpoint, selected by standard AI development with an in-domain real-tumor validation set, may not necessarily be generalized to unseen data, especially for out-domain test set. This limitation arises from the validation set failing to adequately represent corner cases.

2. The best model checkpoint, selected by our strategy with a diverse synthetic-tumor validation set, tends to be generalized well to unseen data. This is because the validation set can cover theoretically infinite examples of possible cancerous tumors across diverse conditions.

3. We introduce a novel continual learning framework. This framework integrates a continuous stream of synthetic data, characterized by diverse data distribution, for both training and validation. Traditional validation sets, constrained by static and limited in-domain real tumors, fall short in such a setting, whereas our synthetic tumors can be dynamically tailored to align with emerging distributions. Importantly, our framework can continuously generate tumors spanning a spectrum of sizes—from small to large—enhancing the detection rate of tumors at their early stages.

Although our study focuses on AI in healthcare, the insight should be pertinent to various imaging applications within the field of computer vision. However, at the time this paper is written, very few studies in computer vision have provided evidence that training *exclusively* on generated synthetic data can match or surpass the performance achieved when trained on real data (Black et al., 2023). In specific applications, integrating synthetic data with real data—essentially acting as data augmentation—has been found empirically to boost AI performance (Mu et al., 2020; Luzi et al., 2022; Azizi et al., 2023; Burg et al., 2023). In this regard, data synthesis—cancerous tumor synthesis in particular—in medical imaging is relatively more successful[1] with specific applications benefiting more from training exclusively on synthetic data than real data.

## 2 RELATED WORK

**The dilemma of validation.** In the field of machine learning, it is customary to use finite, static datasets with a pre-defined data split. While this standard offers a fair benchmark for comparing

---

[1]The greater success of data synthesis in medical imaging (reviewed in §2), compared with computer vision, can be attributed to two factors from our perspective. Firstly, the focus is primarily on synthesizing tumors rather than other components of the human anatomy. Secondly, the synthesis of tumors in 3D medical images is less complex as it does not require considerations for intricate variables such as lighting conditions, pose, and occlusion, which are typical in computer vision tasks.

different AI models, it does not accurately represent real-world learning conditions. Two more realistic scenarios often arise in practice.

- The first scenario is the *small data regime*, commonly observed in medical applications due to constraints like disease prevalence and annotation difficulty (Liu et al., 2022). In such cases, curating an appropriate validation set poses a conundrum. A large validation set would compromise the size of the training set, whereas a small one may not sufficiently estimate the model's performance. Despite its critical importance, this issue has yet to receive adequate attention in the field.

- The second scenario involves dealing with *a stream of data*, in a context of continual learning where the model encounters a continuous flow of new data (Purushwalkam et al., 2022). A finite, static validation set proves unsuitable as it cannot accurately assess the model's capability in processing an extensive and diverse data range. We argue that a validation set—made up of real-world data— might not be needed during the training stage in such situations. Given the vastness of the training data, overfitting can be naturally avoided. Consequently, selecting the last-epoch model checkpoint could be a judicious choice.

**Progresses in data synthesis.** Real-world data often encounters challenges such as poor quality, limited quantity, and inaccessibility. To tackle these obstacles, the notion of *synthetic data* has emerged as a practical alternative, allowing for the generation of samples as needed (Chen et al., 2021). This approach has proven valuable in addressing data limitations and facilitating machine learning processes, including computer vision (Chen et al., 2019; Ramesh et al., 2021), natural language processing (Collobert & Weston, 2008; Brown et al., 2020), voice (Oord et al., 2016), and many other fields (Wiese et al., 2020; Jin et al., 2018; Zheng et al., 2023). In the medical domain, the practice of data synthesis—*tumor synthesis* in particular—endeavors to produce artificial tumors in the image, which can significantly diversify the data and annotations for AI training. Successful works related to tumor synthesis include polyp detection from colonoscopy videos (Shin et al., 2018), COVID-19 detection from Chest CT (Yao et al., 2021; Lyu et al., 2022), diabetic lesion detection from retinal images (Wang et al., 2022), cancer detection from fluorescence microscopy images (Horvath et al., 2022), and brain tumor detection from MRI (Wyatt et al., 2022). A most recent study (Hu et al., 2023) indicated that AI trained *exclusively* on synthetic tumors can segment liver tumors with accuracy comparable to AI trained on real tumors.

To the best of our knowledge, data synthesis has been widely recognized for its contribution in enhancing training and test datasets (Liu et al., 2023). However, its capacity for improving the validation set remains largely untapped. In this paper, we extend the application of synthetic data to the validation set, enabling the full use of the annotated data for AI training while ensuring diverse and comprehensive validation data in the framework of continual learning.

## 3 METHOD & MATERIAL

### 3.1 OUR SETTING FOR CONTINUAL TUMOR SEGMENTATION

According to Van de Ven & Tolias (2019); van de Ven et al. (2022), continual learning can be categorized into three settings: class-incremental learning, task-incremental learning, and domain-incremental learning. In the domain-incremental learning setting, which is relevant to our situation, the problem remains consistent while the context and input distribution change. More specifically, the model sequentially encounters a continuum of input datasets

$$\{X_1, Y_1\}, \{X_2, Y_2\}, ..., \{X_N, Y_N\}, \tag{1}$$

where each dataset $\{X_i, Y_i\}_{1 \leq i \leq N}$ is non-iid dataset. The objective is to train a model $\mathscr{F} : X \rightarrow Y$ that can be effectively queried at any given time to predict the target value $Y$ associated with a test input pair $X$. In liver tumor segmentation settings, $X$ is the input CT volume together with the segmentation mask $Y$. The non-iid refers to those originating from disparate medical centers.

The fixed *static* setting with real data showed in Figure 1(a), where the term *static* denotes the unchanged distribution of the dataset, presents several limitations:

Firstly, the real data is constrained in terms of its scales and acquisition sources. Some datasets contain only dozens of CT volumes sourced from a single medical center (Kavur et al., 2021). This

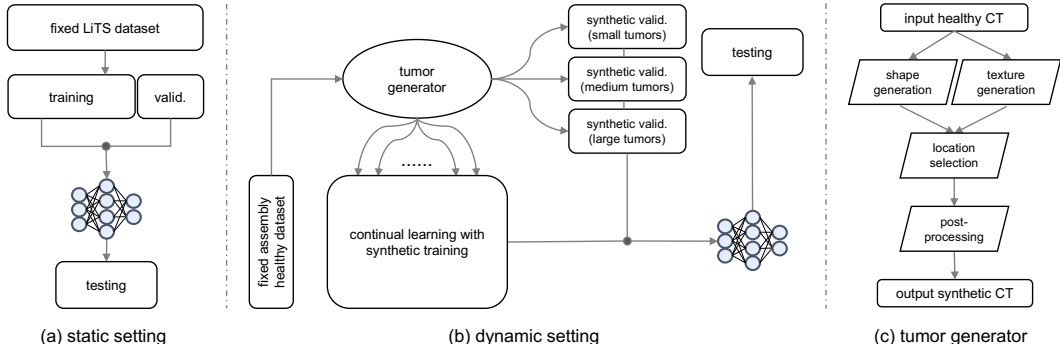

Figure 1: **Illustration of continual learning framework paradigm.** **(a)** The diagram for the fixed static setting, where the real-tumor dataset is partitioned into training and validation sets. The AI model is then developed using these datasets and subsequently tested with unseen data. **(b)** Continual learning framework integrated with synthetic data. **(c)** Tumor generator pipeline. By leveraging an advanced tumor generator, we can create dynamic training set and an extensive validation set. Models developed using dynamic setting exhibit superior performance compared to the fixed static setting.

constraint, combined with the fixed data distribution within the static setting, poses challenges in achieving diversification. Secondly, the task of data annotation, specifically tumor annotation, is exceptionally challenging as it often necessitates the use of corroborative pathology reports. This requirement adds to the difficulty and complexity of extending the dataset. Furthermore, there are specific cases, such as extremely small tumors, where obtaining real data becomes significantly challenging, rendering it impractical or even unattainable. As a result, training and validating the AI model on a fixed static setting is quite likely to result in biased, sub-optimal performance on unseen data, especially for out-domain test sets.

In contrast, the *dynamic* setting shown in Figure 1(b), where the term *dynamic* denotes that the distribution of the dataset changes all the time. We continuously creates a stream of synthetic data for AI training, can overcome the aforementioned limitations:

Firstly, acquiring healthy CT volumes is much easier than obtaining CT volumes with annotated cancerous tumors. As a result, our continual learning framework can start from a diverse dataset comprising CT volumes of healthy subjects procured from multiple sources or medical centers. Secondly, by implementing dynamic control over our pipeline, we gain the ability to generate synthetic data according to specific demands, including those of a tiny radius. Consequently, our framework achieves a noteworthy level of diversity, encompassing a wide array of variations. Therefore, the AI model developed using this continual learning framework of synthetic data has potential to enhance its performance on out-domain data.

### 3.2 MODELING-BASED TUMOR GENERATOR

Inspired by the standardized guidance, medical knowledge, and distributional characteristics of tumors, we develop a modeling-based strategy to generate synthetic data. For example, in liver tumors, according to the Liver Imaging Reporting and Data System (LI-RADS) (Chernyak et al., 2018), the malignancy of hepatocellular carcinomas is determined by shape, size, location and texture, enhancing capsule appearance. The statistical distribution of liver tumors size and intensity can be found in Appendix A. We use a sequence of morphological image-processing operations to model real tumors, as shown in Figure 1(c). The tumor generator consists of four steps: (1) shape generation, (2) texture generation, (3) location selection, and (4) post-processing.

1. **Shape generation.** Based on clinical knowledge, a tumor is initiated from a malignant cell and gradually proliferates and expands, resulting in a nearly spherical shape for small tumors (≤5mm). On the other hand, statistical distributions of real liver tumors indicate that larger tumors tend to exhibit an elliptical shape. This observation has inspired us to generate a tumor-like shape using an ellipsoid $ellip(a, b, c)$, where $a, b, c$ are the length of the semi-axes. Additionally, we utilize elastic deformation (Ronneberger et al., 2015) to enhance the authenticity of the generated tumor

Table 1: **Datasets description.** The LiTS dataset was used to train, validate, and evaluate AI models in segmenting liver tumors. The FLARE'23 dataset was used for external validation. An assembly of the CHAOS, BTCV, and Pancreas-CT datasets was used for generating synthetic training and validation sets, in which the liver in these datasets is confirmed to be healthy. For simplicity, cohorts 1–7 will be referred to throughout the remainder of this paper.

| dataset | notation | split | annotation | # of CTs | tumor |
|---|---|---|---|---|---|
| LiTS (Bilic et al., 2019) | cohort 1 | training | ✓ | 25 | real |
| | cohort 2 | validation | ✓ | 5 | real |
| | cohort 3 | testing | ✓ | 70 | real |
| Assembly (Hu et al., 2023) | cohort 4 | training | ✗ | 25 | synthetic |
| | cohort 5 | validation | ✗ | 50 | synthetic |
| FLARE'23 (Ma et al., 2022) | cohort 6 | validation | ✗ | 50 | synthetic |
| | cohort 7 | testing | ✓ | 120 | real |

    shapes $D(ellip(a, b, c), \sigma_d)$, where $\sigma_d$ control the magnitude of displacements. We show some examples in the Appendix B.

2. **Texture generation.** The generation of textures is a significant challenge due to the varied patterns found in tumors. Our current understanding of tumor textures is derived solely from clinical expertise, which considers factors such as the attenuation value and the distribution characteristics. To achieve the desired texture, we introduce Gaussian noise $\mathcal{N}(\mu, \sigma_g)$ with a predetermined mean attenuation value, matching the standard deviation of the liver tumors. Subsequently, we use cubic interpolation to smooth the texture. Furthermore, to better replicate textures obtained from CT imaging, we use a final step of texture blurring. Examples can be found in Appendix C.

3. **Location selection.** Liver tumors generally do not allow the passage of preexisting blood vessels from the host tissue through them. To address this concern, we initially perform voxel value thresholding for vessel segmentation (Gonzalez, 2009). Utilizing the vessel mask acquired from this step enables us to identify if a particular location can cause tumor-blood collision.

4. **Post-processing.** The post-processing involves evaluating image characteristics through visual inspection and feedback from medical professionals. The purpose of these steps is to replicate the phenomena of mass effect and the appearance of a capsule (Lee et al., 2004). Mass effect refers to the phenomenon wherein the tumor undergoes growth, resulting in the displacement and deformation of surrounding tissues. We utilize local scaling warping (Glasbey & Mardia, 1998) to replicate this effect. Additionally, we brighten the edges of the tumor, thereby simulating the capsule appearance. The output CT volumes with synthetic data are now able to serve the continual learning framework, where you can find some visual examples in Appendix D.

## 4 EXPERIMENT

### 4.1 DATASET & BENCHMARK

Table 1 summarizes a total of five publicly available datasets used in this study. We group them into three classes as follows.

- **Real-tumor dataset.** We select the LiTS dataset (Bilic et al., 2019) for training and testing AI models. LiTS provides detailed per-voxel annotations of liver tumors. The tumor types include HCC and secondary liver tumors and metastasis derived from colorectal, breast, and lung cancer. The size of liver tumors ranges from 38mm$^3$ to 349 cm$^3$, and the radius of tumors is approximately in the range of [2, 44] mm. LiTS is partitioned into a training set (*cohort 1*; 25 CT volumes), validation set (*cohort 2*; 5 CT volumes), and test set (*cohort 3*; 70 CT volumes).

- **Healthy CT assembly.** We have collected a dataset of 75 CT volumes with healthy liver assembled from CHAOS (Kavur et al., 2021), Pancreas-CT (Roth et al., 2016) and BTCV (Landman et al., 2015). This assembled dataset is partitioned into training set (*cohort 4*; 25 CT volumes) and validation set (*cohort 5*; 50 CT volumes). As illustrated in Figure 1(b) For the training set, tumors were dynamically generated within these volumes during training, resulting in a sequential

collection of image-label pairs comprising synthetic tumors. For the validation set, we generated three different tumor sizes (small, medium, and large) for each healthy CT volume offline, giving a total of 150 CT volumes.

- **External benchmark.** FLARE'23 (Ma et al., 2022) is used for external benchmark because it provides out-domain CT volumes from the LiTS dataset. This dataset was specifically chosen due to its extensive coverage, encompassing over 4000 3D CT volumes obtained from more than 30 medical centers. The inclusion of such a diverse dataset ensures the generalizability of the benchmark. The FLARE'23 dataset contains partially labeled annotations. To ensure the suitability of the test set, specific criteria are applied to the annotations. These criteria require that the annotations include per-voxel labeling for both the liver and tumors, with the additional constraint that the connected component of the tumor must intersect with the liver. Adhering to these conditions, we chose the external test set (*cohort 7*; 120 CT volumes). Additionally, same as assembly dataset, we can use the healthy cases within the FLARE'23 to generate synthetic data to serve as *in-domain* validation set (*cohort 6*; 50 CT volumes), which will be used in §5.5.

**Experiment Setup.** Our model is trained for 6,000 epochs, with a model checkpoint being saved every 100 epochs, a total of 60 model checkpoints are saved throughout the entire training process. To ensure robustness and comprehensiveness in obtaining results, the experiment is conducted ten times each to perform statistical analysis. By averaging all runs, we obtain reliable results.

## 4.2 IMPLEMENTATION

We have implemented our codes utilizing the MONAI[2] framework for the U-Net architecture (Ronneberger et al., 2015), a well-established network commonly employed in medical image segmentation tasks. During the pre-processing stage, input images undergo clipping with a window range of [-21,189]. Following this, they are normalized to achieve a zero mean and unit standard deviation (Tang et al., 2022). For training purposes, random patches with dimensions of $96 \times 96 \times 96$ are cropped from the 3D image volumes. A base learning rate of 0.0002 is utilized in the training process, accompanied by a batch size of two per GPU. To further enhance the training process, we employ both the linear warmup strategy and the cosine annealing learning rate schedule. During the inference phase, a sliding window strategy with an overlapping area ratio of 0.75 is adopted. The segmentation performance is evaluated using the Dice Similarity Coefficient (DSC) score, while Sensitivity is used to evaluate the performance of detecting very tiny liver tumors.

## 5 RESULT

### 5.1 THE OVERFITTING IS ATTRIBUTED TO SMALL-SCALE, REAL-TUMOR VALIDATION

To demonstrate the potential limitations of selecting the best checkpoint based on a small-scale and biased real-tumor validation set, we evaluate all the model checkpoints on cohort 2, 3, and 7. Cohort 3 (in-domain test set) assesses the performance of each checkpoint and aids in determining the effectiveness of the selected best checkpoints using the validation set (cohort 2). Cohort 7 (out-domain test set) serves as a robust benchmark, providing an enhanced evaluation of the performance of the model checkpoints on out-domain unseen data.

Figure 2 and Table 2 show the evaluation results. Two significant observations can be made. Firstly, the best checkpoint identified by the small real-tumor validation set exhibits considerable instability, with notable variations observed when different validation samples are chosen. This result indicates that the small-scale real validation is inherently biased and lacks the ability to adequately represent the broader range of cases. Secondly, the performance of the best checkpoint determined by the real validation set does not effectively generalize to unseen test data, particularly when confronted with out-domain data originating from other medical centers. These observations indicate that overfitting can be attributed to a small-scale, biased real-tumor validation set.

---

[2]https://monai.io/

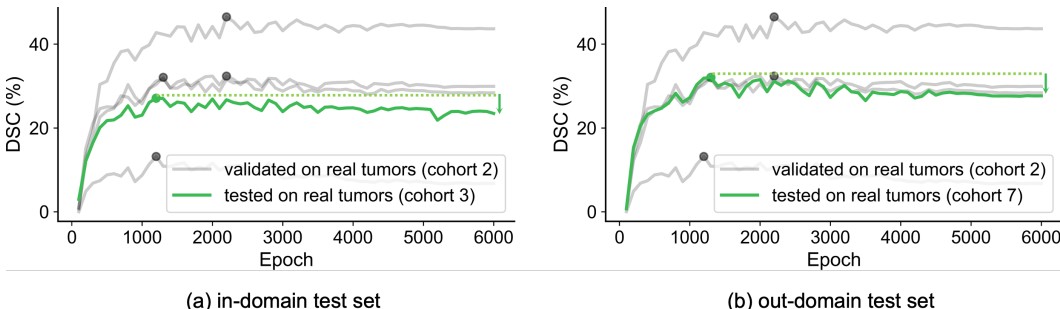

(a) in-domain test set                    (b) out-domain test set

Figure 2: **The overfitting is due to a small-scale, biased real-tumor validation set.** The green curve depicts the performance of each checkpoint on the unseen test set. **(a)** In-domain test set from the LiTS dataset and **(b)** out-domain test set from FLARE'23 dataset. As observed, the model initially performs well, but its performance starts to decline when trained for an extended duration. This decline is attributed to overfitting (green dotted line), where the model becomes too specialized on the training set and loses its ability to generalize effectively to the unseen test set. The purpose of a validation set is to identify the peak performance of the model on unseen data. However, in real-world scenarios, the validation set is often too small, which leads to inaccurate identification of the best checkpoint. The curves generated by the real-tumor validation are plotted in gray. The dots represent the best checkpoint identified by the real-tumor validation (in gray) and determined by the test set (in green). The proximity of the two colored dots relates to the performance of validation set.

## 5.2   THE OVERFITTING IS ALLEVIATED BY LARGE-SCALE, SYNTHETIC-TUMOR VALIDATION

In order to demonstrate the effectiveness of the large-scale and diverse synthetic-tumor validation set in mitigating the issue of overfitting, we conducted a replicated experiment. This experiment involved evaluating all the model checkpoints on cohorts 5, 3, and 7.

The evaluation trajectory can be observed in Figure 3, with the synthetic-tumor validation set represented by the red line. Detailed comparisons with the real-tumor validation set can be found in Table 2. Upon analysis of Table 2, it becomes evident that the best checkpoint selected using the synthetic-tumor validation set demonstrates significantly improved performance compared to the best checkpoint chosen using the real-tumor validation set when tested with unseen data. This enhanced performance is particularly notable when faced with out-domain data (cohort 7). These findings underscore the effectiveness of our diverse and large-scale synthetic-tumor validation set as an improved alternative to address overfitting issues.

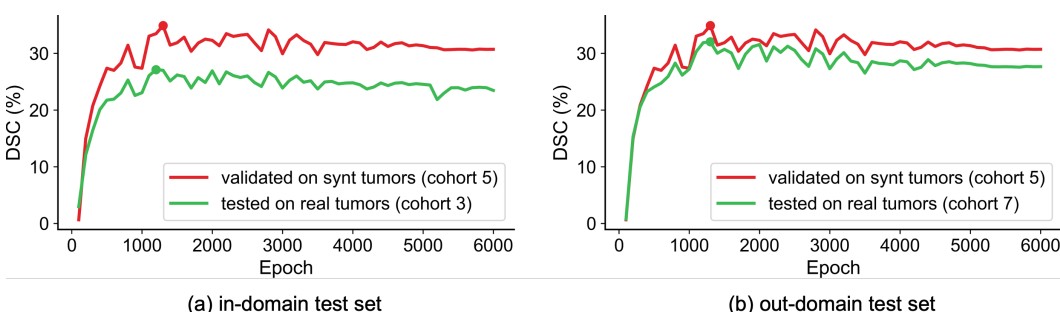

(a) in-domain test set                    (b) out-domain test set

Figure 3: **The overfitting is alleviated by a large-scale, synthetic-tumor validation set.** The red curve represents the results obtained using the synthetic-tumor validation set (cohort 5). **(a-b)** denotes same meaning as Figure 4. We can generate theoretically limitless tumors under diverse conditions using the tumor generator. This extensive coverage enhances our ability to identify the peak performance of the model on unseen datasets. The training trajectory demonstrates the efficacy of the synthetic-tumor validation set, thus highlighting the effectiveness of using a large-scale synthetic-tumor validation set to address the issue of overfitting.

### 5.3 THE OVERFITTING IS ADDRESSED BY CONTINUAL LEARNING ON SYNTHETIC DATA

In the preceding section, we showcased how the utilization of a comprehensive and diverse synthetic-tumor validation set can alleviate the concern of overfitting. Now, we will shift our attention to emphasizing the efficacy of synthetic data in handling the overfitting problem from a training perspective. For this purpose, we compare the performance of an AI model trained on either fixed static real data or dynamic synthetic data.

Figure 4 illustrates the complete training trajectory, while Table 2 provides valuable insights regarding the AI model's performance. Specifically, the AI model trained on fixed static real data demonstrates a DSC of 26.7% for cohort 3 and 31.1% for the out-domain test set cohort 7. In a comparative analysis, the AI model trained on dynamic synthetic data achieves significantly higher DSC values, specifically 34.5% for cohort 3 and 35.4% for cohort 7. These results indicate a notable improvement compared to the AI model trained on real data. Based on these findings, we can confidently assert that incorporating our continual learning framework with synthetic data allows us to effectively address the issue of overfitting, encompassing both the training and validation perspectives.

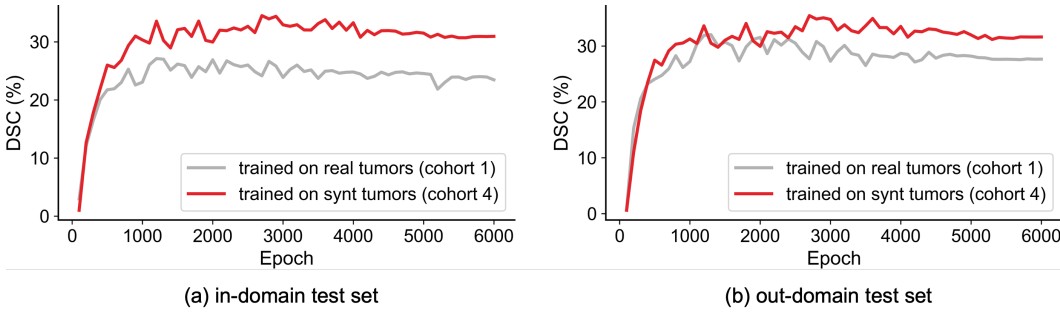

(a) in-domain test set        (b) out-domain test set

Figure 4: **The overfitting is addressed by continual learning on synthetic data. (a)** in- and **(b)** out-domain test set. The gray line corresponds to the AI model trained on fixed, static real data, while the red line represents the model developed through continual learning on synthetic data. It is clear that the AI model trained on synthetic data outperforms the one trained on real data.

Table 2: **Demonstration on liver tumor segmentation.** Cohort 3 and cohort 7 are from the LiTS and FLARE datasets, respectively, detailed in Table 1. The best @ real or @ synt denotes the selection of the *best* checkpoint based on either real- or synthetic- tumor validation set. We report the DSC score (%) (95% CI) achieved on the test set. The result reveals that the dynamic synthetic data demonstrates superior performance than fixed static real data, particularly in out-domain test set.

|  | training on real tumors | | training on synthetic tumors | |
| --- | --- | --- | --- | --- |
|  | best @ real | best @ synt | best @ real | best @ synt |
| cohort 3 | 26.7 (22.6-30.9) | **27.0 (23.7-30.3)** | 33.4 (28.7-38.0) | **34.5 (30.8-38.2)** |
| cohort 7 | 31.1 (26.0-36.2) | **32.0 (28.5-35.5)** | 33.3 (30.6-36.0) | **35.4 (32.1-38.7)** |

### 5.4 SYNTHETIC DATA CAN BENEFIT EARLY CANCER DETECTION

Tiny tumors (radius < 5mm) detection plays a critical role in clinical applications, providing valuable information for early cancer diagnosis. Acquiring real data of such a small size is challenging, often posing difficulties or even making it impossible to acquire them. However, our strategy can dynamically generate numerous tiny tumors as required. As a result, the AI model developed within the continual learning framework yields a significant improvement in detecting tiny liver tumors.

The improvement can be found in Figure 5. We assessed the sensitivity of the AI model under different settings. The performance of the AI model trained and validated on the fixed static setting with real data is 33.1% for the in-domain test set (cohort 3) and 33.9% for the out-domain test set (cohort 7). Comparatively, the AI model developed using our continual learning framework on synthetic data gives a sensitivity of 55.4% for cohort 3 and 52.3% for cohort 7. These results demonstrate the effectiveness of our continual learning framework in early detection of cancer.

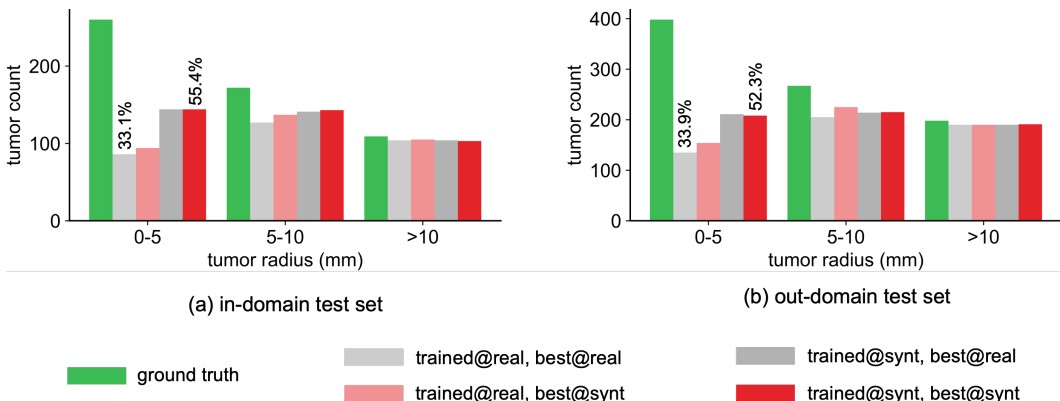

Figure 5: **Synthetic data can benefit early cancer detection.** The trained@real or @synt denotes that the training framework is based on either a fixed static real or dynamic synthetic data set. The best@real or @synt denote the same meaning as Table 1. The numbers on the bar are sensitivities. We show that our continual learning framework is effective in detecting tiny tumors (radius < 5mm).

## 5.5 CONTINUAL LEARNING IS ENHANCED BY IN-DOMAIN SYNTHETIC-TUMOR VALIDATION

We have demonstrated that our continual learning framework yields superior performance compared to fixed static real-tumor datasets. Moving forward, let's consider the continual learning framework itself. Synthetic data offer a significant advantage in that they allow for the utilization of healthy CT volumes from various sources. This means that we can directly incorporate the healthy cases within the dataset and generate an in-domain validation set (cohort 6). As shown in Figure 6, the continual learning framework can lead to more favorable outcomes when we are able to generate in-domain data from the same dataset, as opposed to out-domain synthetic data (cohort 5).

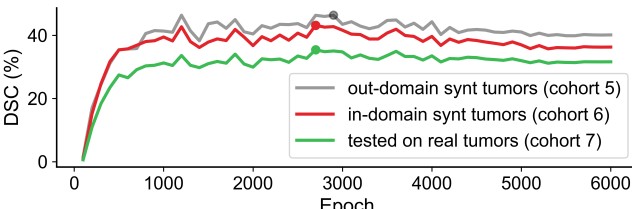

Figure 6: **The continual learning is enhanced by in-domain synthetic-tumor validation.** The gray curve represents the out-domain validation set with synthetic data (cohort 5), while the red curve corresponds to the in-domain set (cohort 6). As evidenced, the performance of our continual learning framework exhibits improvement when utilizing in-domain synthetic data as validation set.

## 6 CONCLUSION

Data synthesis strategies continue to pique the interest of researchers and practitioners, propelling ongoing investigations within this field. This paper justifies the potential and stresses the necessity of leveraging synthetic data as validation to select the best model checkpoint along the training trajectory. Moreover, by employing continual learning framework on synthetic data, we realize a marked improvement in liver tumor segmentation as well as in the early detection of cancerous tumors compared to fixed static setting on real data, where procuring ample annotated examples can be cost-prohibitive. It is particularly valuable in scenarios characterized by limited annotated data.

**Limitation and future work.** The computational cost is high when utilizing offline-generated synthetic data for validation. Additionally, the design and implementation of the tumor generator pipeline pose challenges when attempting to adapt it to different organs, requiring substantial design considerations and specialized expertise. Future investigation could explore more generalized tumor synthesis strategies to facilitate the continual learning framework.

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

# Appendix

## Table of Contents

## A    DISTRIBUTION OF REAL-TUMOR

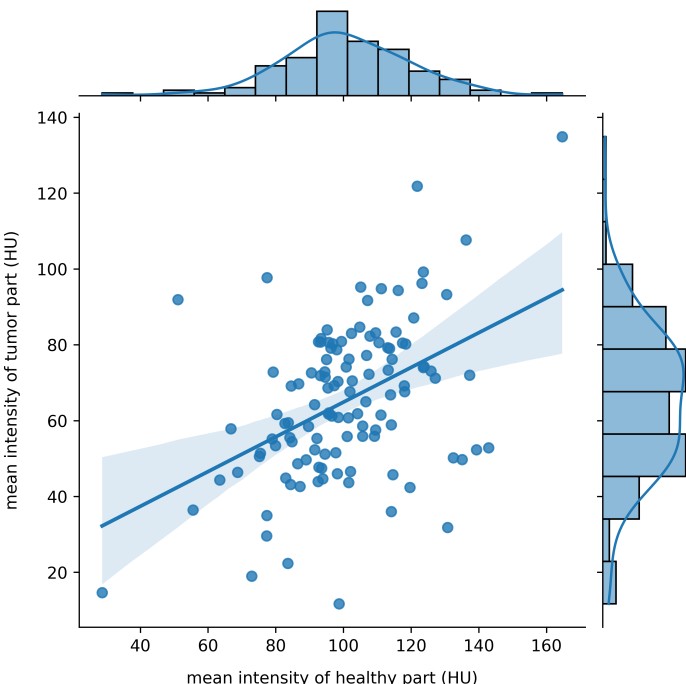

Figure 7: **Intensity distribution of liver tumors and their healthy counterparts.** Our tumor generator, which is based on modeling and medical knowledge, incorporates the distributional characteristics of real tumors. We present the intensity distributions of real tumors obtained from the LiTS dataset.

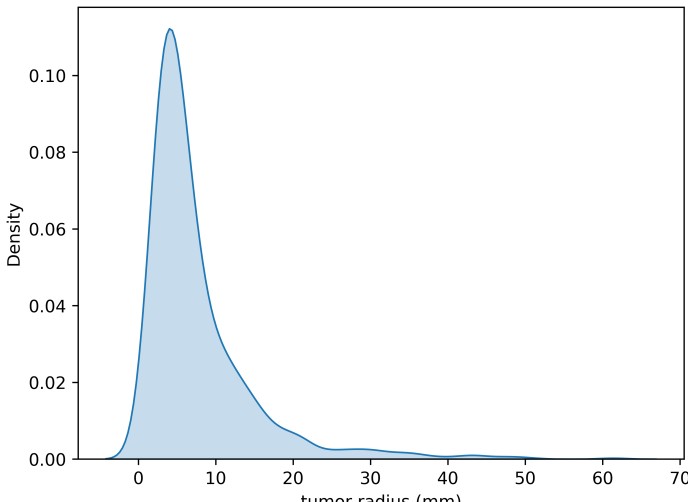

Figure 8: **Size distribution of liver tumors.** We have calculated the size distribution of liver tumors from the LiTS dataset. This tumor size distribution will serve as a guide for determining the sizes of the synthetic tumors we generate.

## B SHAPE EXAMPLES

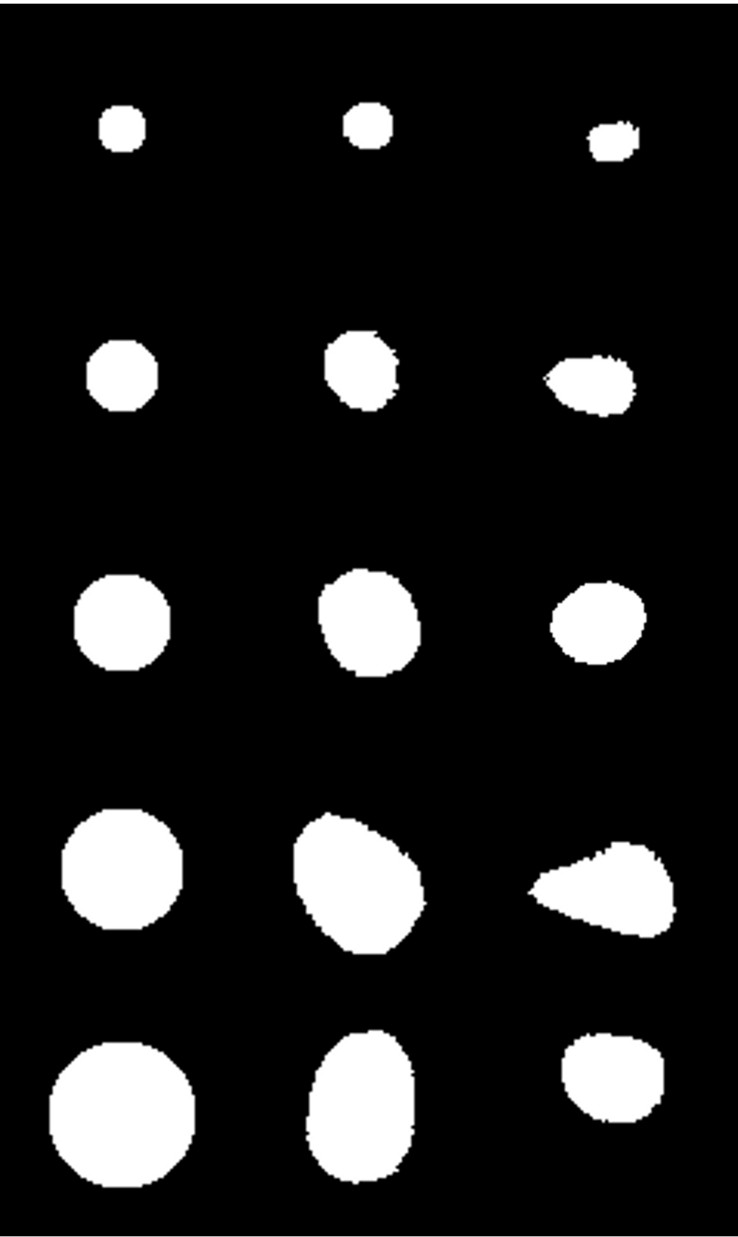

Figure 9: **Examples of the generated shape.** The tumor generator pipeline enables us to control the size and deformation of the generated tumors. Here, we present some examples of generated shapes under different conditions.

## C  TEXTURE EXAMPLES

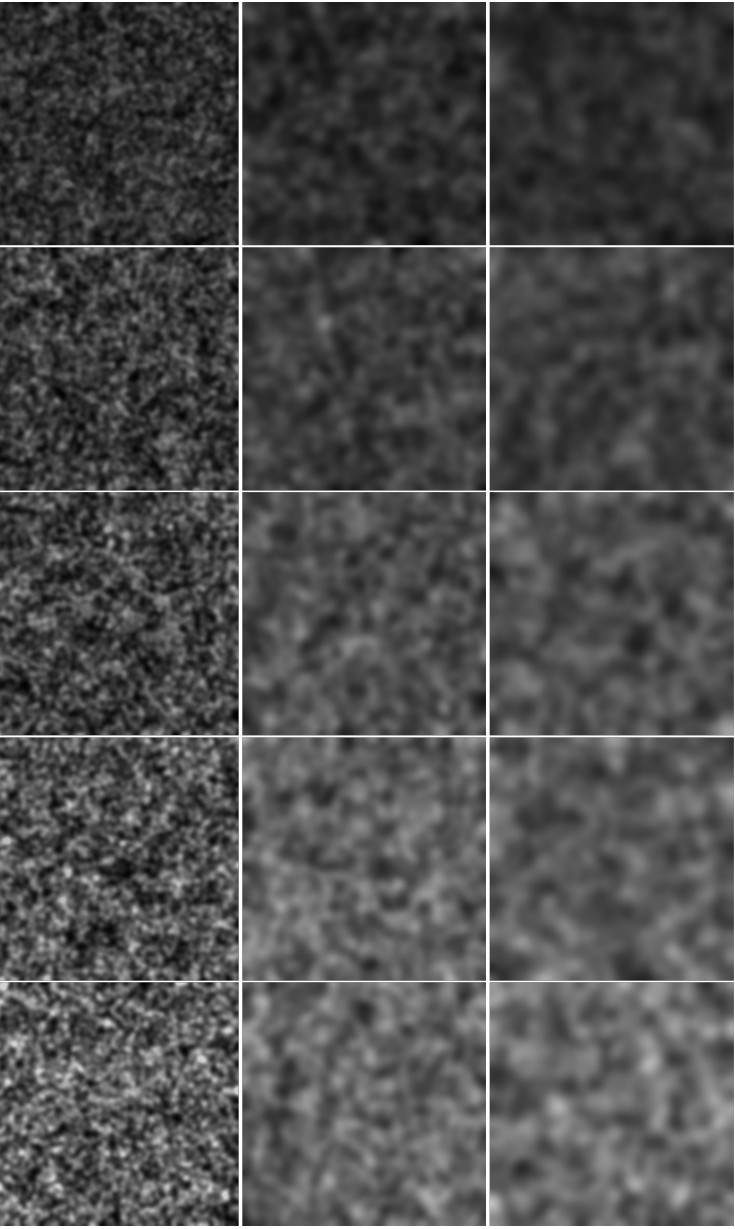

Figure 10: **Examples of the generated texture.** Our data synthesis strategy also enables us to generate different textures, as illustrated here for visualization.

# D  SYNTHETIC TUMOR EXAMPLES

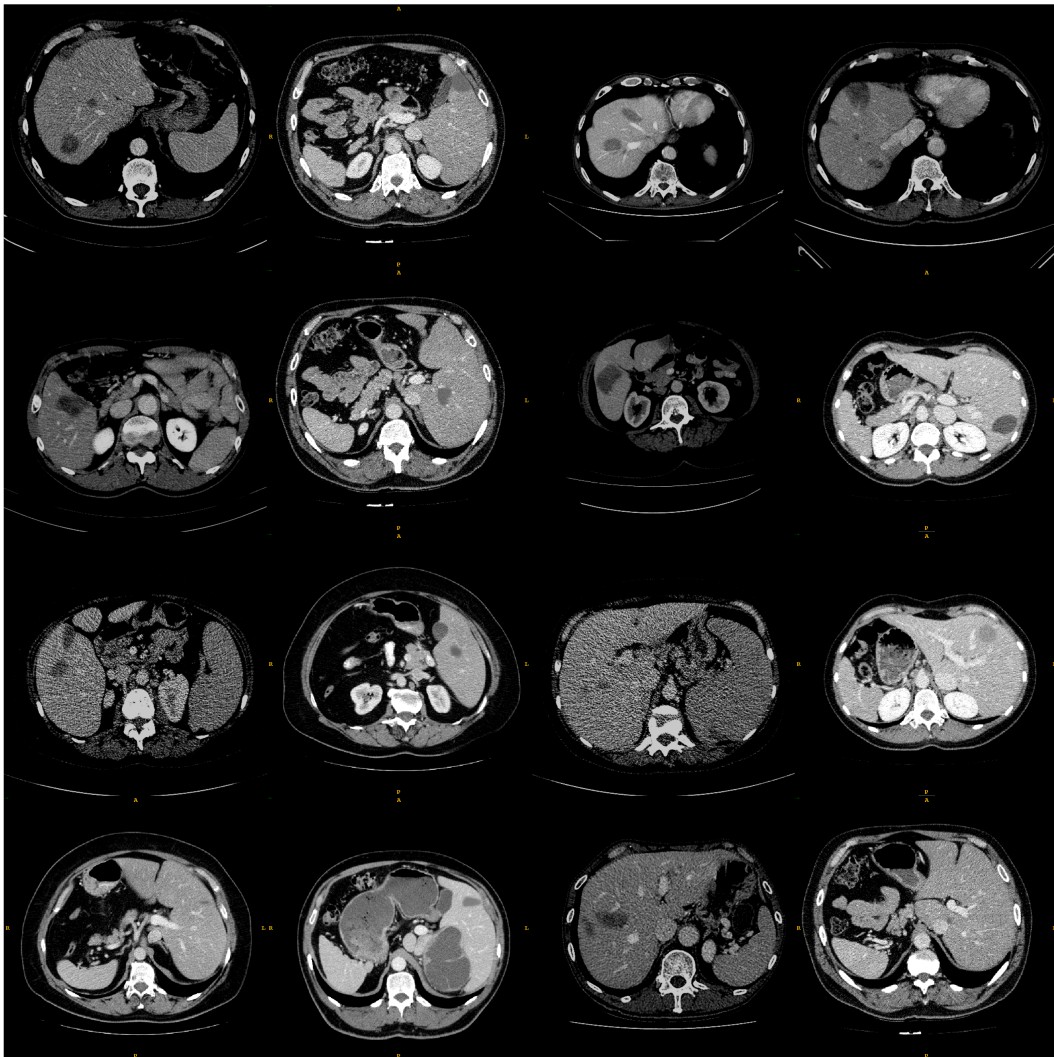

Figure 11: **Visualization of synthetic data.** By combining all the pipelines together, we can obtain a wide range of diverse synthetic data for validation and training.

