# OpenReview forum: "Synthetic Data as Validation"
_ICLR.cc/2024/Conference — Submitted to ICLR 2024_

### Official Review · Reviewer_AHws · 2023-10-12

**Soundness:** 2 fair
**Presentation:** 3 good
**Contribution:** 2 fair
**Rating:** 3
**Confidence:** 4

**Summary:**

While synthetic data have been used for augmenting the training set, this work leverages synthetic data as a validation set to reduce overfitting and ease the selection of the best model in AI development. The authors evaluate it for segmentation in CT volumes, where synthetic tumors are generated to create additional validation data.

**Strengths:**

+ The research question is interesting.
+ Well written paper.
+ The results are well communicated.

**Weaknesses:**

- The authors evaluate synthesising the validation set to decide for checkpoint, which is a nice research question to ask. However, for a fair comparison and to give us readers a better overview, the training set should also be synthesised (with and without synthesising the validation set) and compared.  This will help us understanding which strategy is the best to augment. This is my biggest concern with the paper, since only augmenting the validation set without looking at the training set is not a complete overview.
- See questions below.

**Questions:**

- Appendix D: can you put a rectangle for the generated tumors? This will help reader to spot it easier.
- Appendix D: Can you show real examples of tumors, so that we can compare the synthetic ones and real cases?
- Figure 2: what are the different grey lines? I see that you mention: The curves generated by the real-tumor validation are plotted in gray. But what does each line correspond to, or why do you have multiple grey lines?
- Figure 2: how does the dice score look like for training data?
- Do you have ablation study on number of synthetic validation data? Does it play a role?
- How the tumors are generated seem not to be well studied. Do you have ablation study on this? E.g. is there other ways of texture generation, does the boundary of the tumor play a role, is it always a sharp contrast etc.?
- Section 5.4: how does the tumor count calculated? More information for sensitivity analysis is needed.

Minor:
- Typo on p3: “…We continuously create a stream of synthetic data…” instead of creates.

---

### Official Review · Reviewer_NsC9 · 2023-10-29

**Soundness:** 2 fair
**Presentation:** 2 fair
**Contribution:** 2 fair
**Rating:** 3
**Confidence:** 4

**Summary:**

The paper proposes to use synthetic tumor generation as a means of generating validation data for model selection in liver tumor segmentation. Tumors are inserted into healthy data by a pipeline of shape generation, texture generation, determining location, and post-processing. For experiments, a number of datasets are formulated, with different data sources and tumor generation. Model selection is compared between selection based on a small real validation set and larger synthetic validation sets. Comparisons are also made for training on synthetically generated datasets, supposed to mimic a continual learning strategy.

**Strengths:**

+ The use of synthetic data for validation in limited data scenarios could be a valuable strategy for model selection, and is worthwhile studying closer
+ Continual learning incorporating synthetic data is also a promising direction

More details in questions/comments below.

**Weaknesses:**

- Experiments are limited, with little insight to the impact of validation set size and hyper parameter selection
- Continual learning aspect is unclear, with a fixed synthetic training dataset used for experiments

More details in questions/comments below.

**Questions:**

* The only comparison to validation with real data is a small dataset with 5 volumes, which is difficult to use as a proper baseline. To have actual value of a study on synthetic data for validation, it would be important to understand the interplay between validation set size and test performance. How large val size is needed for robust model selection? How does validation on real vs. synthetic compare at equal validation set size? Would augmentations on real data in validation set be equally valuable compared to adding synthetic data? How about combining real and synthetic data?

* Experiments only test model selection over training length, which requires there to be sufficient variations (e.g. over-fitting) during training. For thorough testing of validation set for model selection, there should also be experiments on a selection of other hyper parameters (e.g. learning rate, architecture selection, augmentation on training data, etc.).

* Figure 2 is the only result supposed to demonstrate the weakness of selection based on the real validation data, but this is rather weak. What are the different curves of real validation? Is this for different models or different volumes ("variations observed when different validation samples are chosen")? Is model selection based on results of single volumes? This does not make sense, and is expected to be very sensitive to the characteristics of individual volumes. Averaging of all validation data should be used for model selection. Also, are these curves averaged over multiple trainings?

* The continual learning setting seems unclear and artificial. For this purpose, a static dataset (cohort 4) is used. In what way does this represent continual learning?

* The results in Fig. 5 mostly demonstrate benefits for training on synthetic data, not validation. As such, these results doesn't contribute to the main aim of the paper of introducing synthetic data for validation.

* Ultimately, the use of synthetic validation data is only tested for one type of data and one synthetic data generation strategy. It is not clear how this would generalize to other tasks, data, and synthetic data generation. As such, the impact of the results is rather weak.

* It would be easier to interpret the generated data in Fig. 11 if segmentation maps were also provided.

* The synthetic tumor generation seems to require quite an amount of manual labor. Would it be possible to use generative modeling for generating tumors, conditioned on, e.g., tumor size and location?

* The term "out-domain" is used throughout the paper, whereas "out-of-domain" or out-of-distribution (OOD) would be more appropriate in relation to the literature.

* It is somewhat confusing reading the introduction, motivating the use of synthetic data for validation, when it suddenly also discusses using synthetic data for training, for example:
"We find that synthetic data can facilitate a more reliable performance estimate on unseen data and effectively address the constraints commonly associated with small, biased validation sets. Specifically, we synthesize tumors in the healthy liver, which gives us orders of magnitude larger datasets for training".
This could be clarified.

---

### Official Review · Reviewer_Nazy · 2023-10-31

**Soundness:** 2 fair
**Presentation:** 4 excellent
**Contribution:** 4 excellent
**Rating:** 3
**Confidence:** 5

**Summary:**

The paper proposed to use synthetic data for ML validation. Specifically, synthetic models of tumors are inserted into the liver region of CT and used to evaluate a (tumor) segmentation model. Three public datasets are split into 7 cohorts of train/validation/test dataset.
The synthetic tumor model generated using statistics based on real data (texture, shape) and inserted into CT of healthy patients with some location consideration (to ensure no collisions with blood vessels).

Experimental results demonstrate that larger synthetic validation sets seem to avoid some of the overfitting issues associated with using a smaller real test set: when the checkpoint is selected using a larger synthetic validation set, results are better than when a smaller real validation set is used. In addition, synthetic data significantly improves performance when used during training.

**Strengths:**

The paper addresses an important problem in medical imaging: lack of large scale real datasets for ML development. In particular, the lack of large-scale and diverse datasets with challenging cases (e.g., for early cancer detection analysis with smaller tumors) make it difficult to create robust AI for such cases. The proposed approach of generating synthetic examples by taking real patient images and inserting synthetic tumors is promising, since generating a full synthetic CT realistically may be more challenging and time consuming.

The paper is well-written and clear. The motivation and related work are very descriptive and convincing.

Paper includes code in the supplementary material (albeit without a readme/instructions).

**Weaknesses:**

There are two key weaknesses.

First, the synthetic tumor model generation is not well-described. It is not clear what dataset are the texture, shape and size characteristics are taken from. Tumors statistics may bias the detection model to detect tumors more easily with from the dataset from which these tumor statistics are obtained from (since the model will learn to search for specific shapes/sizes/textures). This means that only results on an out-of-domain dataset may be indicative of true performance improvement. However, in Figure 5 (b), results with tumor radius 5-10 and >10 are similar across all types of training methods.

Second, dynamically adding synthetic tumors during training can be seen as a data augmentation strategy, but no results are provided with respect to data augmentation (only random patch cropping is used, which takes out information from the images, and may make training more difficult with less realistic inputs). Given the large jump in performance in training on real images to training on synthetic images (compare light gray to dark gray in Figure 5), experiments with data augmentation are needed to confirm that the model is not undertrained (especially in case when trained on real examples).

In addition, results are reported using a single architecture (U-Net), and it is unclear if they will generalize to other architectures. Also, it is not clear what is the resolution of the input CT, and whether small tumors are visible at that resolution. The only qualitative figure provided is Figure 11 in supplementary material, and it is not clear to me where are the tumors in the provided CT images (please add labels/visual guides).

__Other points__:
* Figure 2: there are multiple grey curves, and it is not clear which curve corresponds to the validation set (cohort 2).
* Dice (DSC) is reported with 95% confidence interval (Table 2), but no information about how confidence intervals are computed are provided.
* Authors should consider summarizing cohorts in a table and including type of training cohort within each figure/table for easier reading.

**Questions:**

Please clarify the creation of the tumor model, include guides of where synthetic tumors are present in CT, and provide a comparison of the proposed tumor addition strategy with standard data augmentation methods.

---

### Official Review · Reviewer_j62o · 2023-11-05

**Soundness:** 1 poor
**Presentation:** 2 fair
**Contribution:** 1 poor
**Rating:** 1
**Confidence:** 4

**Summary:**

This paper studies the benefit of using synthetically generated data as a validation set, in the context of medical image segmentation, specifically the case of liver tumor segmentation from computed tomography (CT) data. In the context of this paper, the validation set is used to find the best check point across epochs to select the model (and not, for instance, to select hyperparameters).
	To study the potential benefit of using synthetically generated data as validation set, the authors compare the performances obtained when the validation set is real versus synthetic. Most of their experiments use the Dice similarity coefficient (DSC) as a measure of segmentation accuracy. In addition, they report the voxel-wise sensitivity across different tumor sizes. Furthermore, the authors compare the performances when using real vs synthetic data as training.
	The experiments are performed using a U-Net implemented with MONAI. Several datasets are used and divided into subsets which are used for either as real data training set, real data validation, real data testing set or for synthesizing data for training or validation sets. Synthetic data generation is based on existing work (Hu et al, 2023).

**Strengths:**

1)	To my knowledge, the use of synthetic data as a validation set has not been studied. It looks worthwhile to investigate this, which is the main aim of the present paper.
2)	It is nice that the authors report confidence intervals (at least they do so for the DSC). This is critical to draw robust conclusions from the results and this is unfortunately quite uncommon in the machine learning literature. The authors should be commended for this.
3)	It is a good thing to have two different datasets for testing so that one can study out-of-domain performance.

**Weaknesses:**

1)	A major weakness is that the results do not support the idea that using synthetic data as validation leads to better performance. The main results are those reported in Table 2. In this Table, the use of synthetic data as validation leads to smalls, non-significant, increases in performance (0.3 percent point of DSC for best@synth vs best@real on cohort 3, 0.9 percent point on cohort 7 when training on real data; 1.1 percent point and 2.1 percent points when training on synthetic data, with largely overlapping confidence intervals). On the contrary, the main improvements seem to be obtained when using synthetic data for training, in particular for the in-domain testing (between 6.7 and 7.5 percent points and with confidence intervals which are either minimally overlapping or non-overlapping). However, the benefit of using synthetic data for training is not new and not in line with the title nor the rationale put forward in the introduction of the paper.
2)	Related to the comment above, there are unsubstantiated claims in the paper. In particular, it is not true that *“Upon analysis of Table 2, it becomes evident that the best checkpoint selected using the synthetic-tumor validation set demonstrates significantly improved performance compared to the best checkpoint chosen using the real-tumor validation set when tested with unseen data.”* (see my comment above). Also, it is not demonstrated that *synthetic data can benefit early cancer detection* (see weakness 5 above). This is a very strong clinical claim and the paper does not support it.
3)	The authors experiment with a single, very small, real validation set size (5 samples). It is difficult to draw robust conclusions based on this single analysis. I would have expected to see experiments with varying real validation set sizes. Moreover, using 5 samples as validation seems very small, at least for this application.
4)	The average DSC seems very low. In medical image segmentation, a DSC of about 30% is considered poor. For this task, when looking at the results presented from the LiTS challenge (Bilic et al, 2023, The Liver Tumor Segmentation Benchmark (LiTS), Medical Image Analysis, Volume 84, 2023, 102680), most of the competitors are way above this performance: most above 50%, some reaching 70% and actually only four appear to be around 30% or below. And these are results from 2017 and 2018 challenges, so it could be expected that some progress has been made since then (which seems to be the case when looking at the current leaderboard). Do not get me wrong, I perfectly understand that the focus of the paper is not to improve liver tumor segmentation over the state of the art and thus there is no need to achieve *excellent* performances. Nevertheless, in order to draw robust conclusions, it is needed to have *decent* performances which are in line with typical results reported in the litterature. Otherwise, it may be that the suggested improvements would not hold for approaches that achieve typical performance.
5)	If I understand correctly, Section 5.4 uses **voxel-wise** sensitivity (the percentage of segmented voxels among true positive voxels). If that’s the case, the authors are not assessing early cancer detection. This needs to be replaced with detection metrics. Also, analyzing sensitivity alone is misleading since it does not account for false positives. Overall, the claim that *synthetic data can benefit early cancer detection* seems unsubstantiated.
6)	The continual learning part is not clearly explained. It is not even clear if this is actually continual learning. Is it really the case that synthetic tumors are dynamically tailored to align with emerging distributions? If so, how is this done. Is the model continuously updated with dynamically generate samples or simply retrained with the new samples? Is the problem of forgetting assessed or addressed?

**Questions:**

**Questions**

1)	Could the authors please explain why the DSC is so low (around 30%) and why it is much lower than that of most methods tackling the same task (see Weakness #5 for further explanations)
2)	Could you please explain the rationale for choosing such a very small (5 samples) real validation dataset which is not a realistic setting for many applications and the rationale for not experimenting with varying validation set sizes?
3)	Please clarify if the metric used in Section 5.4 is the “voxel-wise” sensitivity. If that’s the case, it is not adapted to assess detection and should be replaced with “lesion-wise” detection metrics. Also, analyzing sensitivity alone is not sufficient.
4)	Please clarify the continual learning procedure, its dynamic aspect, how synthetic tumors are dynamically tailored to align with emerging distributions and if/how forgetting is assessed/dealt with.
5)	The paper mentions 100 CT volumes for the LiTS dataset while the LiTS challenge website mentions that 130 CT scans are made available to the participants for training. Could you please explain the difference?

**Comments (I added the following comments because I think they may be useful to the authors for improving the paper but they don’t call for a rebuttal).**

1)	Even if the paper were to make a convincing case that using synthetic data as validation leads to improvements (which is not the case currently), the title is too general as the paper only studies this for a particular task and a particular type of data. It is perfectly understandable that it is not possible to study many tasks and different data types in a conference paper but then the paper title should reflect this, e.g. “Using synthetic data as validation set for medical image segmentation”
2)	There are dozens of variations of the U-Net. Please specify which U-Net architecture is used and provide the exact reference (the reference currently given is the original 2D U-net and I doubt this is the architecture that was used in the present paper). I suggest to provide the exact reference and to describe the architecture in details in the appendix.
3)	Please provide confidence intervals for all reported metrics, not only DSC.
4)	I don’t agree that one of the factors, mentioned in footnote 1 page 2, explaining greater utility of data synthesis in medical imaging compared to CV is that “Firstly, the focus is primarily on synthesizing tumors rather than other components of the human anatomy.” On the contrary, I believe that papers such as SynthSeg (Billot et al, 2023) (B Billot, DN Greve, O Puonti, A Thielscher, K Van Leemput, B Fischl, ..., SynthSeg: Segmentation of brain MRI scans of any contrast and resolution without retraining, Medical Image Analysis, 102789) have shown the opposite. On the other hand, the second mentioned factor, although speculative (tumor variability is debatable) seems ok to mention.
5)	I think the work by Billot et al (2023) should be mentioned in the related work as this is a good example of successful training using synthetically generated medical images that produces variations of the anatomy and the image contrast (rather than variation of a lesion such as a tumor).
6)	It seems that the tumor generator described in Section 3.2 is exactly the same as that presented in (Hu et al, 2023) however this is not explicitly said in Section 3.2. The introduction suggests that it is (“we employ a modeling-based strategy (Hu et al., 2023)”) and by comparing this reference to Section 3.2, they seem similar. If the authors are directly using the generator proposed by Hu et al (2023), please state clearly in Section 3.2 that the procedure is identical and that this section is simply a summary of this procedure. On the other hand, if the two are not identical, please explain the differences.
7)	Please clarify that tumor characteristics that serve for the generation come only from the LiTS dataset (Appendix A suggest so but it would be better to be more explicit).
8)	Statistically speaking, I don’t think one can say the real-tumor validation dataset is “biased” (Section 5.1). The problem here is the variability of the sampling process, but the sampling process itself is not biased (it is just that out of the validation samples that may be selected, some may be very far off the testing distribution).
9)	Do not say “significantly” unless it is statistically significant. If you want to say that there is a strong difference, you can say “substantial”.
10)	I suggest to clarify if some samples of LiFT have been left out and why. Also, for reproducibility of the paper, it would be good to include the exact list of samples from each dataset that were included. For FLARE’23, this would have the added benefit to ensure that there was no cherry-picking during the selection.


Minor suggestions

-	Please always say “validation set” and not “validation” since validation can refer to the whole validation procedure (data splits, metrics…)
-	p1, abstract: I don’t think “rigorous” validation is the right wording (since here validation refers to validation set, not to the overall validation procedure). I suggest to replace with “robust” or something like that.
-	“out-of-domain” seems grammatically more correct than “out-domain”
-	I think one should not say “the model… may not be generalized” but rather “the model… may not generalize”
-	Page 4, last paragraph: when referencing Ronneberger et al, 2015, the authors are specifically referring to the elastic deformations used for data augmentation in this paper but this could be made clearer.
-	Figure 7: please define HU: Hounsfield unit
-	Figure 11: please also display the labelled tumor (superimposed on the CT scan) as an additional panel below the image
-	Section 4.2: please mention which loss was used

---

### Official Review · Reviewer_QJkJ · 2023-11-09

**Soundness:** 1 poor
**Presentation:** 2 fair
**Contribution:** 1 poor
**Rating:** 3
**Confidence:** 3

**Summary:**

The authors proposed a strategy of using synthetic data in AI model validation to overcome the issue of limited data in the training set. They generated synthetic liver tumors to demonstrate the performance for early cancer detection in CT volumes. Results were compared using either real or synthetic data in training, validation, and test sets.

**Strengths:**

The authors clearly stated the questions and gaps that need to be addressed and introduced the topic effectively in sections 1 and 2. The question raised is an important one in the field of AI in healthcare. Moreover, the authors provided the code which improves the transparency and reproducibility of their work.

**Weaknesses:**

- Acknowledging myself as a non-expert in the field of tumor image segmentation, I found some of the content unclear and the language vague, needing a more detailed explanation. E.g., in section 3.2, the authors mentioned using a sequence of morphological image-processing operations without providing specific details. Additionally, there is a lack of information on how the authors employed elastic deformation and what the 'D()' is. The texture generation and post-processing sections also need further elaboration and clarity.
- There was a lack of comparison between synthetic data and real data or demonstration of how well the synthetic data mimic real data, or how the proposed tumor generator compared with other tumor simulators.
- Considering the relatively low Dice Similarity Coefficients (DSCs) observed across experiments, the practical utility of this strategy seems to be limited.
- Although the authors attempted to address an important problem, the novelty of the work itself appeared to be constrained, as did the scope of its application without substantial design considerations and specialized expertise.
- As other reviewers mentioned, the main improvement was seen in using synthetic data in training instead of validation, which deviates from the main claim that the authors made.

**Questions:**

- What do texture and intensity imply in the context of tumors?
- Considering that only 120 CT volumes remained, how diverse is cohort 7?
- Apart from tumor size, were there any other differing factors in the healthy CT assembly? Is this cohort representative and diverse?
- In Figure 2, why are there multiple lines for validation? Were different validation samples selected, and if so, how?
- In Figure 3, it is challenging to make a direct comparison visually with Figure 2. Unsure which experiment corresponds to which number in Table 2. What data were used for training here?
- In Figure 5, sensitivity was shown but did the authors look at false positives?
- In Figure 6, it appears that the out-domain had a higher DSC than the in-domain. Were the test results were based on in-domain or out-domain synthetic-tumor validation?
- How high is the computational cost?

---

### Meta-Review · Area_Chair_jdkZ · 2023-12-07

**Metareview:**

The reviewers raised the issue that the manuscript makes claims but the evidence is lacking. In particular, there is no evidence that synthetic data benefits early-stage detection of cancer and overall fragile experimental results.

**Justification For Why Not Higher Score:**

Overall, the results are not convincing.

**Justification For Why Not Lower Score:**

N/A

---

### Decision · Program_Chairs · 2024-01-16

Reject